# Diagnostic Age, Age at Death and Stage Migration in Men Dying with or from Prostate Cancer in Denmark

**DOI:** 10.3390/diagnostics12051271

**Published:** 2022-05-19

**Authors:** Marc Casper Meineche Andersen, Hein Vincent Stroomberg, Klaus Brasso, John Thomas Helgstrand, Andreas Røder

**Affiliations:** 1Copenhagen Prostate Cancer Center, Department of Urology, Copenhagen University Hospital, Rigshospitalet, 2200 Copenhagen, Denmark; hein.vincent.stroomberg@regionh.dk (H.V.S.); klaus.brasso@regionh.dk (K.B.); john.thomas.autrup.helgstrand@regionh.dk (J.T.H.); andreas.roeder@regionh.dk (A.R.); 2Department of Clinical Medicine, University of Copenhagen, 2200 Copenhagen, Denmark

**Keywords:** prostatic neoplasms, stage migration, clinical characteristics, cause of death, Danish Prostate Cancer Registry (DaPCaR)

## Abstract

The impact of changes in diagnostic activity and treatment options on prostate cancer epidemiology remains a subject of debate. Newly published long-term survival outcomes may not represent contemporary patients and new perspectives are in demand. All men dying in Denmark with prostate cancer diagnosis during a 10-year period were analyzed to address the stage migration of and time lived with prostate cancer diagnosis. All male deaths in Denmark between 2007 and 2016 (*n* = 261,657) were obtained and crosslinked with The Danish Prostate Cancer Registry (DaPCaR) and the Danish Cancer Registry. Correlation in diagnostic age and stage (localized, locally advanced, metastatic), age at death and cause of death were investigated by Kruskal-Wallis test and linear regression in 15,692 men diagnosed with prostate cancer. Prostate cancer mortality remained stable during the study period. Among the men who died of prostate cancer, 65% had locally advanced or metastatic disease at diagnosis. Age at diagnosis declined in men diagnosed with localized disease and remained constant in men with locally advanced or metastatic disease. Age at death increased in all men. Despite increased efforts to detect prostate cancer early, two-thirds of men who die from prostate cancer still have advanced prostate cancer at the time of diagnosis. Our data show increased life-expectancy in men diagnosed with prostate cancer, however, this benefit must be weighed against increased time of living with the disease and overdiagnosis. The intensified treatment of elderly men and men with advanced disease may be the key to lower prostate cancer mortality.

## 1. Introduction

Prostate cancer has a long natural history and extensive follow-up is needed to determine disease specific outcomes [1,2,3]. Time lived with prostate cancer can be prolonged by earlier diagnosis and/or prolonged life due to better treatment options, as illustrated in Figure 1. Screening with prostate-specific antigen (PSA), whether opportunistic or systematic, has been estimated to decrease age at diagnosis in the range of 3–12 years [4,5,6]. Furthermore, the use of more sensitive diagnostics, finding smaller metastatic lesions, may introduce stage migration and a Will-Rogers phenomenon [7]. Epidemiological studies of prostate cancer survival are vulnerable to these biases due to the slow growing nature of the disease and they potentially overestimate the gain in survival over time.

In Denmark, prostate cancer incidence has increased in the past 20 years without a reduction in prostate cancer mortality [8]. It is important to understand how age at diagnosis and age at death have changed over the years, and especially how these changes relate to the stage at diagnosis, as this reflects the current diagnostic strategy that is primarily driven by the use of PSA as a tool for early detection. Recent studies have shown that most men dying of prostate cancer have advanced disease at diagnosis [9,10]. Despite a very high age-standardized incidence rate in Denmark, this suggests that the current diagnostic strategy has not eliminated metastatic disease and the introduction of curative treatment has not decreased mortality [8].

In this paper, all Danish men who died during a 10-year period with a previous prostate cancer diagnosis were studied and temporal changes in age at diagnosis and age at death were analyzed. Moreover, trends in stage at diagnosis stratified for cause of death was investigated.

## 2. Materials and Methods

All men who died between 1 January 2007 and 31 December 2016 in Denmark were identified in the Danish Registry of Causes of Death (RCOD). The RCOD contains information on the date and underlying cause of death classified by the International Classification of Diseases 10th edition (ICD-10) [11]. ”Prostate cancer-specific death” was assigned if primary cause of death was either “DC61” or “DC61.9”. All other causes of death were defined as “other cause death”.

Information on prostate cancer diagnosis and clinical characteristics were extracted from the Danish Prostate Cancer Registry (DaPCaR), a national registry of all men who have had prostate tissue pathologically examined in Denmark since 1995, regardless of the final pathological assessment [12]. Additional clinical information was integrated from the Danish Cancer Registry (DCR), containing information on all cancers diagnosed in Denmark since 1943 [13]. If DaPCaR had recorded a T-category of cTx, T-category from DCR was acquired if available, and similarly for both N- and M-category. All information was linked by the Danish civil registration number, a unique number given to every Danish citizen [14]. Before 2004, the DCR classified all cancers according to the World Health Organization (WHO) classifications as localized, regionally advanced, or distant metastatic. Since 2004, the tumor, lymph node, and metastases (TNM) classification of malignant tumors has been used. To make clinical stages uniform, we classified men with cT1-T2, N0/x, M0/x as localized, cT3-T4 and/or N1, M0/x as locally advanced and any M1 disease as metastatic, according to previous methodology [9]. Men were excluded if stage was not assessed, i.e., if TNM-category was Tx/Nx/Mx or missing, or if diagnosis was based on autopsy findings.

Linear regression models were used to investigate the per-year change in observed age at diagnosis and age at death accompanied by a 95% confidence interval (95CI). The linear regression model was used to predict the age at diagnosis and age at death in men who died between 2007 and 2016 to calculate the time from diagnosis to death. Trends in proportion of prostate cancer deaths and in distribution of stage at diagnosis were calculated using Kruskal-Wallis test. Statistical significance was defined as *p*-values less than 0.05. Data management and analysis were performed using R/Rstudio version 1.4.1106.

The registry was approved by the Danish Data Protection Agency (file number: 2012-41-0390), the Research Ethics Committee of the Capital Region of Denmark (local journal number: VD-2019-22), and the Danish Patient Safety Authority with reference 3-3013-2861/1.

## 3. Results

During the studied period, 261,657 men died; among these 38,687 had prostate evaluations in the DaPCaR, and 19,615 had a prostate cancer diagnosis, and 3923 men were excluded based on the exclusion criteria. The final cohort for analysis consisted of 15,692 men as shown in Figure 2. Basic characteristics for men with prostate cancer-specific and men with other cause death is shown in Table 1. In total, 7751 had localized, 4076 had locally advanced, and 3865 men had metastatic prostate cancer at the time of diagnosis. The median prostate-specific antigen (PSA) level at diagnosis was 30.1 ng/mL (inter quartile range (IQR): 12-103), with PSA values missing for 6666 men. 3197 men were diagnosed with a Gleason score (GS) of 6 or below; 4594 were diagnosed with a GS of 7; 7353 were diagnosed with a GS of 8 or above; and 548 were diagnosed with an unspecified adenocarcinoma, neuro endocrine or small cell carcinoma. Curative intended radical prostatectomy had been performed in 737 men diagnosed with localized disease and 8325 prostate cancer-specific deaths were observed in the cohort.

Figure 3A illustrates the changes in the mean age at diagnosis and death in the entire cohort. Age at diagnosis significantly decreased at a rate of 0.12 years-per-year (95CI: 0.07–0.16, *p* < 0.001), and age at death increased by 0.23 years-per-year (95CI: 0.18–0.27, *p* < 0.001). The predicted time from diagnosis to death increased from 3.1 years in 2007 to 6.2 years in 2016. Figure 3B shows the distribution of the cause of death in men with prostate cancer. The proportion of prostate cancer-specific death in men with a prostate cancer diagnosis decreased from 732 out of 1179 men in 2007 to 889 out of 1837 men in 2016 (*p* < 0.001).

Figure 3C illustrates the change in mean age at diagnosis and death stratified by stage at diagnosis. Age at diagnosis decreased by 0.29 years-per-year (95CI: 0.23–0.35, *p* < 0.001) in men diagnosed with localized disease while the age at diagnosis remained constant in men diagnosed with locally advanced (0.03 years-per-year, 95CI: −0.07–0.13, *p* = 0.54) or metastatic disease (0.02 years-per-year, 95CI: −0.08–0.12, *p* = 0.64). Age at death increased by 0.09 years-per-year (95CI: 0.03–0.15, *p* = 0.004), 0.36 years-per-year (95CI: 0.27–0.46, *p* < 0.001), and 0.23 years-per-year (95CI: 0.13–0.33, *p* < 0.001) for men diagnosed with localized, locally advanced or metastatic disease, respectively. Figure 3D shows the distribution of the cause of death stratified by stage at diagnosis. Men diagnosed with metastatic disease had a high proportion of prostate cancer-specific death (76.5%), which remained constant over time (*p* = 0.59). The proportion of prostate cancer-specific death decreased over time in men diagnosed with localized disease from 46.9% to 31.5% and (*p* < 0.001) and with locally advanced disease from 66.2% to 59.2% (*p* = 0.003).

Figure 4A–C illustrates the distribution of diagnostic stage among all men who died and stratified for prostate cancer-specific death and other cause death. Among the 8325 men that died of prostate cancer 2958 were diagnosed with localized disease and the proportion of diagnostic stage remained constant over time (*p* = 0.24). In men dying from other causes than prostate cancer, 2574 men out of 7367 were diagnosed with non-localized disease and the proportion of the diagnostic stage remained constant over time (*p* = 0.096). The proportion of men with locally advanced prostate cancer increased from 141 out of 732 in 2007 to 322 out of 889 in 2016 (*p* < 0.001) and from 72 out of 447 to 216 out of 948 in 2016 (*p* = 0.002) in men who died of prostate cancer or of non-prostate cancer-related causes, respectively.

Figure 4D illustrates the change in mean age at diagnosis and death stratified by stage at diagnosis and cause of death. Age at death in men with localized disease dying of prostate cancer increased at a rate of 0.23 years-per-year (95CI = 0.14–0.34, *p* < 0.001) while age at diagnosis decreased by 0.1 years-per-year (95CI = 0.00–0.20, *p* = 0.049). Men with localized disease dying of other causes showed no change in age at death (0.03 years-per-year (95CI = −0.06–0.10, *p* = 0.54) while the age at diagnosis decreased noticeably at a rate of 0.39 years-per-year (95CI = 0.31–0.47, *p* < 0.001). In men with locally advanced or metastatic disease at diagnosis, age at diagnosis did not change and age at death increased when stratified by cause of death, Appendix A.

## 4. Discussion

Early detection of cancer has long been promoted as the best approach to reduce prostate cancer mortality. The current strategy for finding prostate cancer early includes digital rectal examination, transrectal ultrasound, and biomarkers [1,3,15,16,17]. However, PSA remains the only well described biomarker for early disease detection, despite PSA not being cancer specific. Even though PSA screening reduces the incidence of metastatic prostate cancer by almost 50% [18], it reduces prostate cancer mortality to a limited extent [19]. Therefore, PSA screening is currently not implemented in Denmark, despite no implementation of screening Denmark has witnessed a marked increase in diagnostic activity [20,21,22]. Previous studies have indicated that opportunistic screening accounts for approximately 16% of the PSA tests taken [21]. Therefore, epidemiological studies of prostate cancer are needed to investigate the effects of our prostate cancer detection strategy. To our knowledge, this study is the first that has backtracked all dead men over ten years to study prostate cancer epidemiology from a new perspective.

Overall, we found that among all Danish men who died during the studied period, 4.5% had prostate cancer recorded as their primary cause of death, corresponding to the expected prostate cancer mortality in Denmark, which has been nearly constant over a longer period [8,11,23]. Almost 65% of men who died from prostate cancer had advanced disease at diagnosis. In men with advanced disease at diagnosis, age at diagnosis remained unchanged, but age at prostate cancer specific and other cause death increased, leading to an overall increase in the age at death of almost 2 years. This increase in age of death is likely a result of better treatment options including taxane-based chemotherapy and new hormonal agents [24,25]. However, increased age of death may also be caused by stage migration, as due to the increased sensitivity and use of advanced imaging more men with apparently localized prostate cancer will be diagnosed with low burden advanced disease. As men with low burden advanced disease have better prognosis than men with high burden advanced disease, the inclusion of low burden advanced disease will thus lead to a higher mean age at death [7,10,26]. Others have demonstrated similar findings showing an improved survival in men with newly diagnosed metastatic prostate cancer depending on the year of diagnosis in a population-based setting, and, that it was possible to identify more men with low-volume metastatic disease with the use of prostate-specific membrane antigen positron emission tomography compared to conventional imaging [27,28]. These studies point to the fact that both improved treatment and stage migration have contributed to increased survival in men with metastatic prostate cancer. It is surprising that age at diagnosis did not decline in men diagnosed with either locally advanced or metastatic disease during the studied period. This indicates that an earlier diagnosis may not be possible in these patients. Increased survival is therefore likely explained not by earlier diagnosis, but by increased sensitivity for diagnosing low burden metastatic disease [23].

It was further observed that age at diagnosis declined in men diagnosed with localized prostate cancer. This decline is likely a result of increased diagnostic activity in younger men, due to increased interest in curative therapy since its introduction in Denmark in 1995 [23]. In men diagnosed with localized disease, increased overall survival was also observed. This was mostly due to patients being diagnosed younger and not due to a large increase in age at death, indicating a lead-time effect. The finding of only a small increase in age at death is in accordance with previous studies on the length of life after radical prostatectomy showing a minimal post-surgical gain in life-expectancy [29]. Moreover, the increase in age at death corresponds to changes in life-expectancy in the Danish male population [30]. When stratified by cause of death we found that the age at death among men dying of prostate cancer increased at a similar rate across all disease stages at diagnosis. Men diagnosed with localized disease and dying of prostate cancer have most likely progressed and subsequently been treated accordingly; however, this could not be confirmed as information on what treatments were offered was not available. Age at death in men with localized prostate cancer dying from other causes remained stable. Age at death is naturally influenced by both comorbidity and the treatment modality of prostate cancer. Unfortunately, this study lacks information on comorbidity and further research must be conducted.

The current diagnostic strategy in Denmark, driven by unsystematic PSA testing, does not find more advanced prostate cancer earlier which could in part explain why we have not observed a decline in prostate cancer mortality. However, there are more men diagnosed with early stage prostate cancer, as shown by an increased proportion of men dying with a localized prostate cancer diagnosis. However, this did not lead to a reduction in the proportion of men dying of prostate cancer with localized disease indicating that there is an increasing amount of men diagnosed who would have died of other causes regardless of diagnosis. It must be noted that the Danish strategy towards the treatment of localized prostate cancer has historically been rather conservative. Guidelines in the early 2000′s only recommended curative treatment to men with at least 10 years remaining life-expectancy, consequently no treatment was offered to men above 70 years of age. The mean age at diagnosis in men with localized disease was close to 74 years and these men would not have been considered candidates for curative treatment. Therefore, the proportion of men undergoing subsequent radical prostatectomy was low compared to other cohorts [2,22]. These treatment recommendations remain unchanged, although the upper age-limit is gradually increasing as life-expectancy is increasing. Thus, in the context of a conservative approach to prostate cancer treatment in Denmark during the studied period, it is likely, that the older men with localized disease have been prone to under-treatment. The results may, therefore, not be comparable to countries where curative treatment is offered to elderly men as well.

It must be addressed that no decrease was observed in the number of men diagnosed with advanced prostate cancer, which contributed the most to the prostate cancer mortality. A more aggressive strategy in these men could have postponed the time to prostate cancer death. A recent Swedish study indicates that many elderly men with advanced disease may be undertreated, and a change towards a more aggressive treatment policy in these men would likely decrease mortality, which potentially explains the observed decreased mortality in Sweden [31,32]. Lessons from recent studies in both newly diagnosed metastatic and castration-resistant prostate cancer has taught us that multimodal treatment can result in a dramatic improvement in survival compared to a conservative approach with castration-based therapy alone [24,33,34]. Our findings indicate that this aggressive treatment of advanced or advancing disease following the diagnosis of localized prostate cancer played a role in prolonging life following diagnosis.

One of the strengths of this study was that of having complete information on the stage at diagnosis. DaPCaR is a national registry based on the mandatory reporting of histopathological examinations and thus covers all men with histologically verified prostate cancer in Denmark and is regarded as complete. Moreover, the completeness of registry data in Denmark made it possible to have a uniquely large cohort of men dying with prostate cancer diagnosis.

A limitation of our study is that the risk of having prostate cancer recorded as the primary cause of death may be overestimated in registries, as previous studies have demonstrated a difference in the percentage of men dying from prostate cancer depending on whether register data were used, or whether chart reviews were performed [35]. A major limitation is the lacking information on comorbidity and treatments offered. Especially, Since the increasing age at death in patients diagnosed with localized disease corresponds to the expected increase in life-expectancy of the Danish male population, indicating that comorbidity in men included likely were as expected, and because the treatment of prostate cancer has rapidly developed in recent decades, it should therefore be a focus in future research. Furthermore, information on histopathological grading was not sufficiently detailed, which means we cannot take into account the aggressiveness of the disease at diagnosis. Lastly, men with missing clinical stage was evenly distributed throughout the period indicating a general problem in the registries, and were therefore excluded, due to the even distribution of missing data it is not likely that the exclusion affects the analysis of the data.

## 5. Conclusions

Despite increased diagnostic activity to detect prostate cancer early, two-thirds of men who die from prostate cancer still have advanced prostate cancer already at the time of diagnosis. Stage migration was only observed from metastatic to locally advanced disease and questions whether the intense diagnostic activity prevents the lethality of the cancer. The age at diagnosis only decreased significantly in men with localized prostate cancer and substantially in men who did not die of the disease. Our data does show an increased life-expectancy in men diagnosed with prostate cancer, however, this benefit must be weighed against the increased time of living with the disease and overdiagnosis. The aggressive treatment of advanced prostate cancer may have a better chance of changing the epidemiology of lethal prostate cancer, and the potential undertreatment of elderly men with advanced prostate cancer should be investigated further.

## Figures and Tables

**Figure 1 diagnostics-12-01271-f001:**
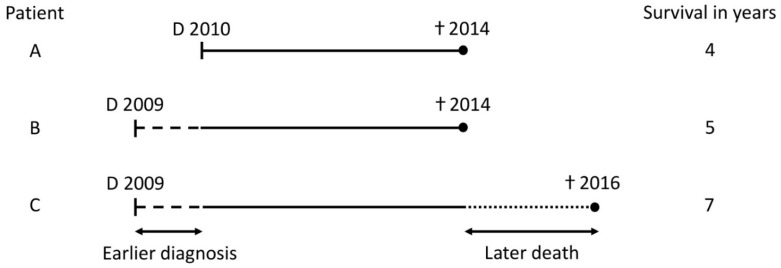
Hypothetical situations for increased survival in cancer patients. †: death; D: diagnosis.

**Figure 2 diagnostics-12-01271-f002:**
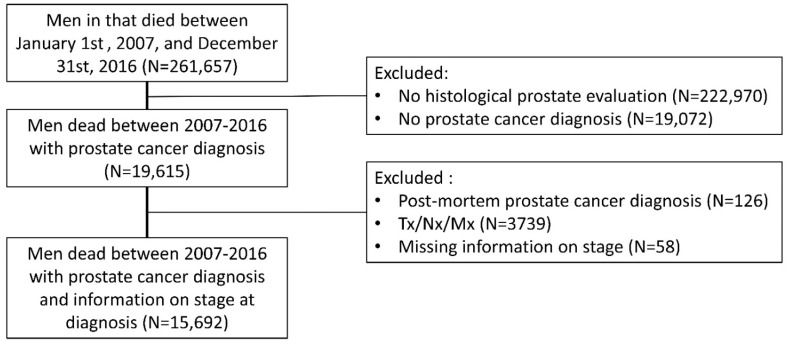
Flowchart of inclusion.

**Figure 3 diagnostics-12-01271-f003:**
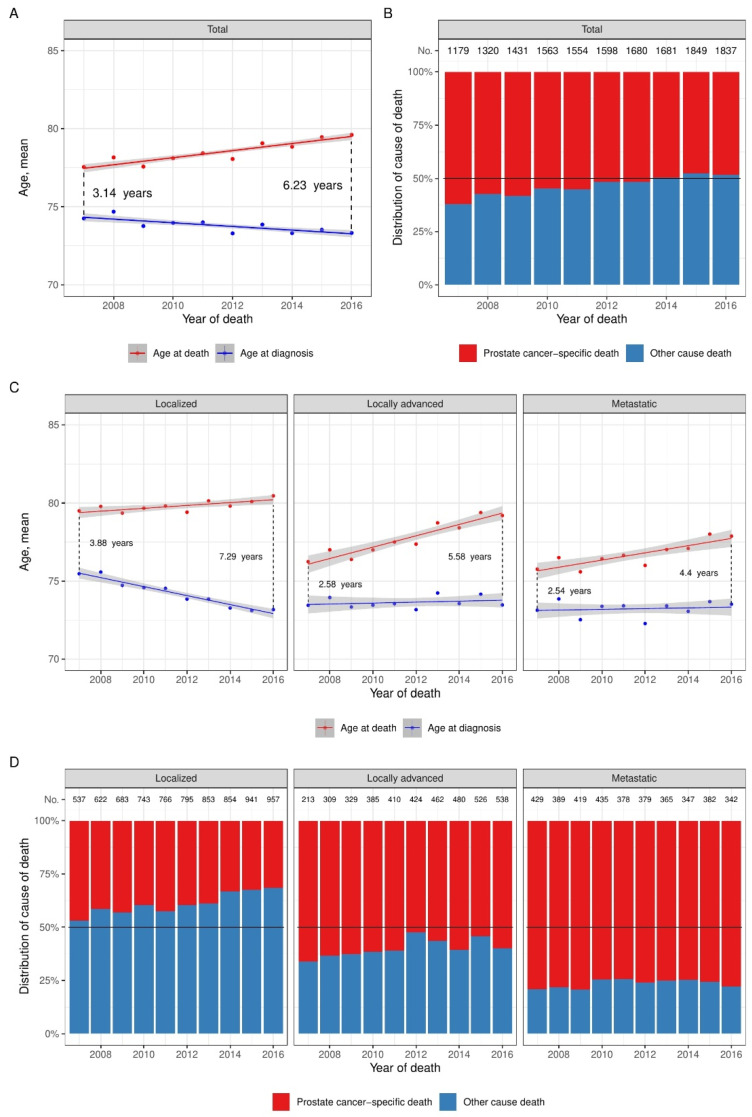
Temporal trends in age at diagnosis, age at death and distribution of cause of death among dead men with prostate cancer, total cohort (*n* = 15,692). (**A**) Linear regression of observed age at death and diagnosis with in grey the 95% confidence interval (95CI) of the regression, and the mean age at diagnosis and mean age at death per year of death in the total cohort. Differences in time from diagnosis to death are calculated as the difference in the predicted age from the linear regression models, respectively; (**B**) Overall distribution of cause of death per year of death. The total number of patients in corresponding year is depicted above each bar; (**C**) Linear regression of observed age at death and diagnosis with the 95CI of the regression in grey, and the mean age at diagnosis and mean age at death per year of death stratified by stage at diagnosis. Differences in time from diagnosis to death are calculated as the difference in the predicted age from the linear regression models, respectively; (**D**) Overall distribution of cause of death per year of death. The total number of patients in the corresponding year is depicted above each bar, abbreviated by “No”.

**Figure 4 diagnostics-12-01271-f004:**
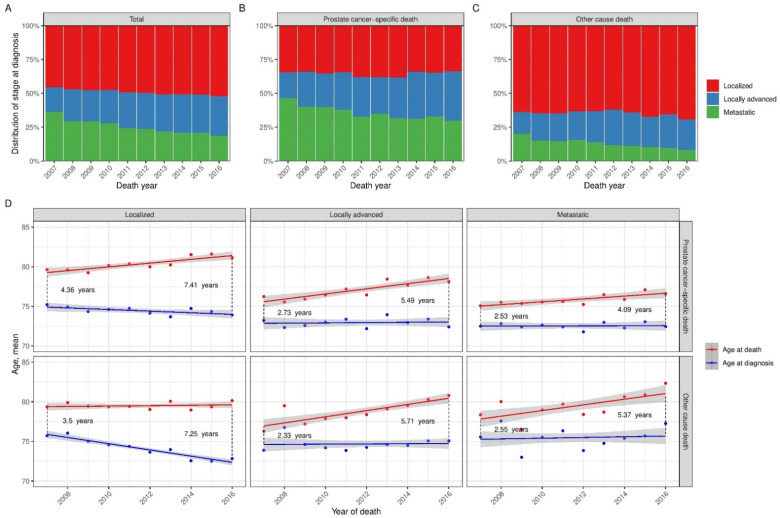
Temporal trends in distribution of stage at diagnosis, age at diagnosis and age at death among dead men with prostate cancer. (**A**) Distribution of stage at diagnosis in total cohort; (**B**) Distribution of stage at diagnosis in men dying of prostate cancer; (**C**) Distribution of stage at diagnosis in men dying of other causes; (**D**) Linear regression of observed age at death and diagnosis with the 95% confidence interval of the regression in grey, and the mean age at diagnosis and mean age at death per year of death stratified by stage at diagnosis and cause of death. Differences in time from diagnosis to death are calculated as the difference in the predicted age from the linear regression models, respectively.

**Table 1 diagnostics-12-01271-t001:** Basic clinical characteristics of included patients.

Variable	Men with Prostate Cancer-Specific Death (*n* = 8325)	Men with Other Cause Death (*n* = 7367)
PSA at diagnosis (ng/mL) ^1^	54 (18–193)	18 (8.8–45)
Missing (number of men)	3513	3153
Diagnostic stage ^2^		
Localized	2958 (35.5%)	4793 (65.1%)
Locally advanced	2411 (29.0%)	1665 (22.6%)
Metastatic	2956 (35.5%)	909 (12.3%)
Gleason score ^2^		
6 or below	966 (11.6%)	2231 (30.3%)
7	2235 (26.8%)	2359 (32.0%)
8 or above	4850 (58.3%)	2503 (34.0%)
Other *	274 (3.3%)	274 (3.7%)
Radical prostatectomy ^2^	213 (2.6%)	591 (8.0%)

^1^ Presented as median (inter quartile range); ^2^ Presented as number of men (percentage of total); * Other includes, unspecified adeno, neuro endocrine and small cell carcinoma. Abbreviation: PSA = prostate-specific antigen.

## Data Availability

The data presented in this study are available on request from the corresponding author. The data are not publicly available due to local guidelines of data storage.

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
