# Peer review of "Diagnostic Age, Age at Death and Stage Migration in Men Dying with or from Prostate Cancer in Denmark"

_diagnostics, 2022, doi:10.3390/diagnostics12051271_

Round 1
Reviewer 1 Report
Thank you for asking me to review the manuscript.
Herewith I am giving my comments. Hope these comments will be helpful for the authors.
Major Essential Revisions
・I can't find the results of survival in abstract. If your focus is age at death, the title was not appropriate.
Did you show data of life-expectancy in your results?
・Please show the characteristics of the men.
I recommend that you make the table of the characteristics of the men.
・The results section (lines 90-92 ) is different in a number of Figure2.
・What do you show in “No” of Figure 3B and Figure 3D?
Please add annotation to the Figure 3B and Figure 3D.
・A numerical formula of Linear regression models is not clear.
Please show a numerical formula of Linear regression models.
Minor Essential Revisions
It is better to align the number of digits after the decimal point.
For example, p=0.24 (line140), p=0.1 (line142), p=0.002 (line144)
Reviewer 2 Report
No suggestions
Author Response
We thank the reviewer for the time spend on reviewing the manuscript.

Round 2
Reviewer 1 Report
No suggestions